# Time-Sensitive Recommendation From Recurrent User Activities

**Nan Du$^{\diamond}$, Yichen Wang$^{\diamond}$, Niao He$^{*}$, Le Song$^{\diamond}$**
$^{\diamond}$College of Computing, Georgia Tech
$^{*}$H. Milton Stewart School of Industrial & System Engineering, Georgia Tech
dunan@gatech.edu, yichen.wang@gatech.edu, nhe6@gatech.edu
lsong@cc.gatech.edu

## Abstract

By making personalized suggestions, a recommender system is playing a crucial role in improving the engagement of users in modern web-services. However, most recommendation algorithms do not explicitly take into account the temporal behavior and the recurrent activities of users. Two central but less explored questions are how to recommend the most desirable item *at the right moment*, and how to predict *the next returning time* of a user to a service. To address these questions, we propose a novel framework which connects self-exciting point processes and low-rank models to capture the recurrent temporal patterns in a large collection of user-item consumption pairs. We show that the parameters of the model can be estimated via a convex optimization, and furthermore, we develop an efficient algorithm that maintains $O(1/\epsilon)$ convergence rate, scales up to problems with millions of user-item pairs and hundreds of millions of temporal events. Compared to other state-of-the-arts in both synthetic and real datasets, our model achieves superb predictive performance in the two time-sensitive recommendation tasks. Finally, we point out that our formulation can incorporate other extra context information of users, such as profile, textual and spatial features.

## 1 Introduction

Delivering personalized user experiences is believed to play a crucial role in the long-term engagement of users to modern web-services [26]. For example, making recommendations on proper items at the right moment can make personal assistant services on mainstream mobile platforms more competitive and usable, since people tend to have different activities depending on the temporal/spatial contexts such as morning vs. evening, weekdays vs. weekend (see for example Figure 1(a)). Unfortunately, most existing recommendation techniques are mainly optimized at predicting users' one-time preference (often denoted by integer ratings) on items, while users' continuously time-varying preferences remain largely under explored.

Besides, traditional user feedback signals (*e.g.* user-item ratings, click-through-rates, *etc.*) have been increasingly argued to be ineffective to represent real engagement of users due to the sparseness and nosiness of the data [26]. The *temporal patterns* at which users return to the services (items) thus becomes a more relevant metric to evaluate their satisfactions [12]. Furthermore, successful predictions of the returning time not only allows a service to keep track of the evolving user preferences, but also helps a service provider to improve their marketing strategies. For most web companies, if we can predict when users will come back next, we could make ads bidding more economic, allowing marketers to bid on time slots. After all, marketers need not blindly bid all time slots indiscriminately. In the context of modern electronic health record data, patients may have several diseases that have complicated dependencies on each other shown at the bottom of Figure 1(a). The occurrence of one disease could trigger the progression of another. Predicting the returning time on certain disease can effectively help doctors to take proactive steps to reduce the potential risks. However, since most models in literature are particularly optimized for predicting ratings [16, 23, 15, 3, 25, 13, 21],

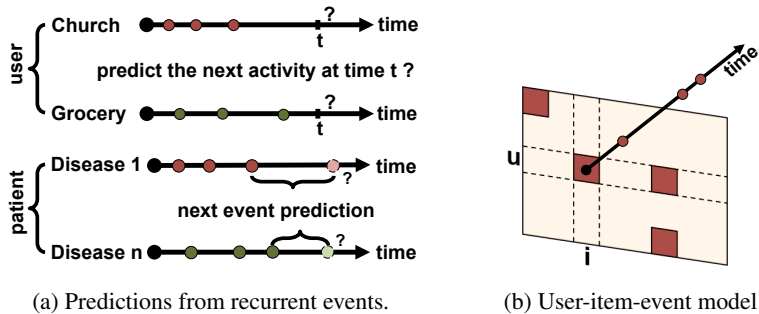

(a) Predictions from recurrent events.          (b) User-item-event model.

Figure 1: Time-sensitive recommendation. (a) in the top figure, one wants to predict the most desirable activity at a given time $t$ for a user; in the bottom figure, one wants to predict the returning time to a particular disease of a patient. (b) The sequence of events induced from each user-item pair $(u, i)$ is modeled as a temporal point process along time.

exploring the recurrent temporal dynamics of users' returning behaviors over time becomes more imperative and meaningful than ever before.

Although the aforementioned applications come from different domains, we seek to capture them in a unified framework by addressing the following two related questions: (1) *how to recommend the most relevant item at the right moment*, and (2) *how to accurately predict the next returning-time of users to existing services*. More specifically, we propose a novel convex formulation of the problems by establishing an under explored connection between self-exciting point processes and low-rank models. We also develop a new optimization algorithm to solve the low rank point process estimation problem efficiently. Our algorithm blends proximal gradient and conditional gradient methods, and achieves the optimal $O(1/t)$ convergence rate. As further demonstrated by our numerical experiments, the algorithm scales up to millions of user-item pairs and hundreds of millions of temporal events, and achieves superb predictive performance on the two time-sensitive problems on both synthetic and real datasets. Furthermore, our model can be readily generalized to incorporate other contextual information by making the intensity function explicitly depend on the additional spatial, textual, categorical, and user profile information.

**Related Work**. The very recent work of Kapoor et al. [12, 11] is most related to our approach. They attempt to predict the returning time for music streaming service based on survival analysis [1] and hidden semi-markov model. Although these methods explicitly consider the temporal dynamics of user-item pairs, a major limitation is that the models cannot generalize to recommend any new item in future time, which is a crucial difference compared to our approach. Moreover, survival analysis is often suitable for modeling a single terminal event [1], such as infection and death, by assuming that the inter-event time to be independent. However, in many cases this assumption might not hold.

## 2 Background on Temporal Point Processes

This section introduces necessary concepts from the theory of temporal point processes [4, 5, 6]. A temporal point process is a random process of which the realization is a sequence of events $\{t_i\}$ with $t_i \in \mathbb{R}^+$ and $i \in \mathbb{Z}^+$ abstracted as points on the time line. Let the history $\mathcal{T}$ be the list of event time $\{t_1, t_2, \ldots, t_n\}$ up to but not including the current time $t$. An important way to characterize temporal point processes is via the conditional intensity function, which is the stochastic model for the next event time given all previous events. Within a small window $[t, t + dt)$, $\lambda(t)dt = \mathbb{P}\{\text{event in } [t, t + dt)|\mathcal{T}\}$ is the probability for the occurrence of a new event given the history $\mathcal{T}$.

The functional form of the intensity $\lambda(t)$ is designed to capture the phenomena of interests [1]. For instance, a homogeneous Poisson process has a constant intensity over time, *i.e.*, $\lambda(t) = \lambda_0 \geqslant 0$, which is independent of the history $\mathcal{T}$. The inter-event gap thus conforms to the exponential distribution with the mean being $1/\lambda_0$. Alternatively, for an inhomogeneous Poisson process, its intensity function is also assumed to be independent of the history $\mathcal{T}$ but can be a simple function of time, *i.e.*, $\lambda(t) = g(t) \geqslant 0$. Given a sequence of events $\mathcal{T} = \{t_1, \ldots, t_n\}$, for any $t > t_n$, we characterize the conditional probability that no event happens during $[t_n, t)$ and the conditional density $f(t|\mathcal{T})$ that an event occurs at time $t$ as $S(t|\mathcal{T}) = \exp\left(-\int_{t_n}^{t} \lambda(\tau)\, d\tau\right)$ and $f(t|\mathcal{T}) =$

$\lambda(t) \, S(t|\mathcal{T})$ [1]. Then given a sequence of events $\mathcal{T} = \{t_1, \dots, t_n\}$, we express its likelihood by

$$\ell(\{t_1, \dots, t_n\}) = \prod_{t_i \in \mathcal{T}} \lambda(t_i) \cdot \exp\left(-\int_0^T \lambda(\tau) \, d\tau\right). \tag{1}$$

## 3 Low Rank Hawkes Processes

In this section, we present our model in terms of low-rank self-exciting *Hawkes* processes, discuss its possible extensions and provide solutions to our proposed time-sensitive recommendation problems.

### 3.1 Modeling Recurrent User Activities with Hawkes Processes

Figure 1(b) highlights the basic setting of our model. For each observed user-item pair $(u, i)$, we model the occurrences of user $u$'s past consumption events on item $i$ as a self-exciting *Hawkes* process [10] with the intensity:

$$\lambda(t) = \gamma_0 + \alpha \sum_{t_i \in \mathcal{T}} \gamma(t, t_i), \tag{2}$$

where $\gamma(t, t_i) \geqslant 0$ is the triggering kernel capturing temporal dependencies, $\alpha \geqslant 0$ scales the magnitude of the influence of each past event, $\gamma_0 \geqslant 0$ is a baseline intensity, and the summation of the kernel terms is history dependent and thus a stochastic process by itself.

We have a twofold rationale behind this modeling choice. First, the baseline intensity $\gamma_0$ captures users' inherent and long-term preferences to items, regardless of the history. Second, the triggering kernel $\gamma(t, t_i)$ quantifies how the influence from each past event evolves over time, which makes the intensity function depend on the history $\mathcal{T}$. Thus, a *Hawkes* process is essentially a conditional Poisson process [14] in the sense that conditioned on the history $\mathcal{T}$, the Hawkes process is a Poisson process formed by the superposition of a background homogeneous Poisson process with the intensity $\gamma_0$ and a set of inhomogeneous Poisson processes with the intensity $\gamma(t, t_i)$. However, because the events in the past can affect the occurrence of the events in future, the Hawkes process in general is more expressive than a Poisson process, which makes it particularly useful for modeling repeated activities by keeping a balance between the long and the short term aspects of users' preferences.

### 3.2 Transferring Knowledge with Low Rank Models

So far, we have shown modeling a sequence of events from a single user-item pair. Since we cannot observe the events from all user-item pairs, the next step is to transfer the learned knowledge to unobserved pairs. Given $m$ users and $n$ items, we represent the intensity function between user $u$ and item $i$ as $\lambda^{u,i}(t) = \lambda_0^{u,i} + \alpha^{u,i} \sum_{t_j^{u,i} \in \mathcal{T}^{u,i}} \gamma(t, t_j^{u,i})$, where $\lambda_0^{u,i}$ and $\alpha^{u,i}$ are the $(u, i)$-th entry of the $m$-by-$n$ non-negative base intensity matrix $\mathbf{\Lambda}_0$ and the self-exciting matrix $\mathbf{A}$, respectively.

However, the two matrices of coefficients $\mathbf{\Lambda}_0$ and $\mathbf{A}$ contain too many parameters. Since it is often believed that users' behaviors and items' attributes can be categorized into a limited number of prototypical types, we assume that $\mathbf{\Lambda}_0$ and $\mathbf{A}$ have low-rank structures. That is, the nuclear norms of these parameter matrices are small $\|\mathbf{\Lambda}_0\|_* \leqslant \lambda', \|\mathbf{A}\|_* \leqslant \beta'$. Some researchers also explicitly assume that the two matrices factorize into products of low rank factors. Here we assume the above nuclear norm constraints in order to obtain convex parameter estimation procedures later.

### 3.3 Triggering Kernel Parametrization and Extensions

Because it is only required that the triggering kernel should be nonnegative and bounded, feature $\boldsymbol{\psi}^{u,i}$ in 3 often has analytic forms when $\gamma(t, t_j^{u,i})$ belongs to many flexible parametric families, such as the Weibull and Log-logistic distributions [1]. For the simplest case, $\gamma(t, t_j^{u,i})$ takes the exponential form $\gamma(t, t_j^{u,i}) = \exp(-(t - t_j^{u,i})/\sigma)$. Alternatively, we can make the intensity function $\lambda^{u,i}(t)$ depend on other additional context information associated with each event. For instance, we can make the base intensity $\mathbf{\Lambda}_0$ depend on user-profiles and item-contents [9, 7]. We might also extend $\mathbf{\Lambda}_0$ and $\mathbf{A}$ into tensors to incorporate the location information. Furthermore, we can even learn the triggering kernel directly using nonparametric methods [8, 30]. Without loss of generality, we stick with the exponential form in later sections.

### 3.4 Time-Sensitive Recommendation

Once we have learned $\mathbf{\Lambda}_0$ and $\mathbf{A}$, we are ready to solve our proposed problems as follows :

(a) **Item recommendation**. At any given time $t$, for each user-item pair $(u, i)$, because the intensity function $\lambda^{u,i}(t)$ indicates the tendency that user $u$ will consume item $i$ at time $t$, for each user $u$, we recommend the proper items by the following procedures :

> 1. Calculate $\lambda^{u,i}(t)$ for each item $i$.
> 2. Sort the items by the descending order of $\lambda^{u,i}(t)$.
> 3. Return the top-$k$ items.

(b) **Returning-time prediction**: for each user-item pair $(u, i)$, the intensity function $\lambda^{u,i}(t)$ dominates the point patterns along time. Given the history $\mathcal{T}^{u,i} = \{t_1, t_2, \ldots, t_n\}$, we calculate the density of the next event time by $f(t|\mathcal{T}^{u,i}) = \lambda^{u,i}(t) \exp\left(-\int_{t_n}^{t} \lambda^{u,i}(t) dt\right)$, so we can use the expectation to predict the next event. Unfortunately, this expectation often does not have analytic forms due to the complexity of $\lambda^{u,i}(t)$ for Hawkes process, so we approximate the returning-time as following :

> 1. Draw samples $\{t_{n+1}^1, \ldots, t_{n+1}^m\} \sim f(t|\mathcal{T}^{u,i})$ by Ogata's thinning algorithm [19].
> 2. Estimate the returning-time by the sample average $\frac{1}{m}\sum_{i=1}^{m} t_{n+1}^i$

## 4 Parameter Estimation

Having presented our model, in this section, we develop a new algorithm which blends proximal gradient and conditional gradient methods to learn the model efficiently.

### 4.1 Convex Formulation

Let $\mathcal{T}^{u,i}$ be the set of events induced between $u$ and $i$. We express the log-likelihood of observing each sequence $\mathcal{T}^{u,i}$ based on Equation 1 as :

$$\ell\left(\mathcal{T}^{u,i}|\mathbf{\Lambda}_0, \mathbf{A}\right) = \sum_{t_j^{u,i} \in \mathcal{T}^{u,i}} \log(\mathbf{w}_{u,i}^\top \boldsymbol{\phi}_j^{u,i}) - \mathbf{w}_{u,i}^\top \boldsymbol{\psi}^{u,i}, \tag{3}$$

where $\mathbf{w}_{u,i} = (\mathbf{\Lambda}_0(u,i), \mathbf{A}(u,i))^\top$, $\boldsymbol{\phi}_j^{u,i} = (1, \sum_{t_k^{u,i} < t_j^{u,i}} \gamma(t_j^{u,i}, t_k^{u,i}))^\top$ and $\boldsymbol{\psi}^{u,i} = (T, \sum_{t_j^{u,i} \in \mathcal{T}^{u,i}} \int_{t_j^{u,i}}^{T} \gamma(t, t_j^{u,i}) dt)^\top$. When $\gamma(t, t_j^{u,i})$ is the exponential kernel, $\boldsymbol{\psi}^{u,i}$ can be expressed as $\boldsymbol{\psi}^{u,i} = (T, \sum_{t_j^{u,i} \in \mathcal{T}^{u,i}} \sigma(1 - \exp(-(T - t_j^{u,i})/\sigma)))^\top$. Then, the log-likelihood of observing all event sequences $\mathcal{O} = \{\mathcal{T}^{u,i}\}_{u,i}$ is simply a summation of each individual term by $\ell(\mathcal{O}) = \sum_{\mathcal{T}^{u,i} \in \mathcal{O}} \ell(\mathcal{T}^{u,i})$. Finally, we can have the following convex formulation :

$$\text{OPT} = \min_{\mathbf{\Lambda}_0, \mathbf{A}} -\frac{1}{|\mathcal{O}|} \sum_{\mathcal{T}^{u,i} \in \mathcal{O}} \ell\left(\mathcal{T}^{u,i}|\mathbf{\Lambda}_0, \mathbf{A}\right) + \lambda\|\mathbf{\Lambda}_0\|_* + \beta\|\mathbf{A}\|_* \text{ subject to } \mathbf{\Lambda}_0, \mathbf{A} \geqslant \mathbf{0}, \tag{4}$$

where the matrix nuclear norm $\|\cdot\|_*$, which is a summation of all singular values, is commonly used as a convex surrogate for the matrix rank function [24]. One off-the-shelf solution to 4 is proposed in [29] based on ADMM. However, the algorithm in [29] requires, at each iteration, a full SVD for computing the proximal operator, which is often prohibitive with large matrices. Alternatively, we might turn to more efficient conditional gradient algorithms [28], which require instead, the much cheaper linear minimization oracles. However, the non-negativity constraints in our problem prevent the linear minimization from having a simple analytical solution.

### 4.2 Alternative Formulation

The difficulty of directly solving the original formulation 4 is caused by the fact that the nonnegative constraints are entangled with the non-smooth nuclear norm penalty. To address this challenge, we approximate 4 using a simple penalty method. Specifically, given $\rho > 0$, we arrive at the next formulation 5 by introducing two auxiliary variables $\mathbf{Z}_1$ and $\mathbf{Z}_2$ with some penalty function, such as the squared Frobenius norm.

$$\widehat{\text{OPT}} = \min_{\mathbf{\Lambda}_0, \mathbf{A}, \mathbf{Z}_1, \mathbf{Z}_2} -\frac{1}{|\mathcal{O}|} \sum_{\mathcal{T}^{u,i} \in \mathcal{O}} \ell\left(\mathcal{T}^{u,i}|\mathbf{\Lambda}_0, \mathbf{A}\right) + \lambda\|\mathbf{Z}_1\|_* + \beta\|\mathbf{Z}_2\|_* + \rho\|\mathbf{\Lambda}_0 - \mathbf{Z}_1\|_F^2$$

$$+ \rho\|\mathbf{A} - \mathbf{Z}_2\|_F^2 \quad \text{subject to} \quad \mathbf{\Lambda}_0, \mathbf{A} \geqslant \mathbf{0}. \tag{5}$$

| **Algorithm 1:** Learning Hawkes-Recommender | **Algorithm 2:** $\mathrm{Prox}_{\boldsymbol{U}^{k-1}}\left(\eta_k \nabla_1(f(\boldsymbol{U}^{k-1}))\right)$ |
|---|---|
| **Input**: $\mathcal{O} = \left\{\mathcal{T}^{u,i}\right\}, \rho > 0$ <br> **Output**: $\boldsymbol{Y}_1 = [\boldsymbol{\Lambda}_0; \boldsymbol{A}]$ <br> Choose to initialize $\boldsymbol{X}_1^0$ and $\boldsymbol{X}_2^0 = \boldsymbol{X}_1^0$ ; <br> Set $\boldsymbol{Y}^0 = \boldsymbol{X}^0$; | $\boldsymbol{X}_1^k = \left(\boldsymbol{U}^{k-1} - \eta_k \nabla_1(f(\boldsymbol{U}^{k-1}))\right)_+$ ; |
| **for** $k = 1, 2, \ldots$ **do** | **Algorithm 3:** $\mathrm{LMO}_\psi\left(\nabla_2(f(\boldsymbol{U}^{k-1}))\right)$ |
| $\quad \delta^k = \frac{2}{k+1}$; <br> $\quad \boldsymbol{U}^{k-1} = (1-\delta^k)\boldsymbol{Y}^{k-1} + \delta^k \boldsymbol{X}^{k-1}$ ; <br> $\quad \boldsymbol{X}_1^k = \mathrm{Prox}_{\boldsymbol{U}^{k-1}}\left(\eta_k \nabla_1(f(\boldsymbol{U}^{k-1}))\right)$; <br> $\quad \boldsymbol{X}_2^k = \mathrm{LMO}_\psi\left(\nabla_2(f(\boldsymbol{U}^{k-1}))\right)$; <br> $\quad \boldsymbol{Y}^k = (1-\delta^k)\boldsymbol{Y}^{k-1} + \delta^k \boldsymbol{X}^k$; <br> **end** | $(u_1, v_1), (u_2, v_2)$ top singular vector pairs of <br> $-\nabla_2(f(\boldsymbol{U}^{k-1}))[\boldsymbol{Z}_1]$ and $-\nabla_2(f(\boldsymbol{U}^{k-1}))[\boldsymbol{Z}_2]$; <br> $\boldsymbol{X}_2^k[\boldsymbol{Z}_1] = u_1 v_1^\top, \boldsymbol{X}_2^k[\boldsymbol{Z}_2] = u_2 v_2^\top$ ; <br> Find $\alpha_1^k$ and $\alpha_2^k$ by solving (6); <br> $\boldsymbol{X}_2^k[\boldsymbol{Z}_1] = \alpha_1^k \boldsymbol{X}_2^k[\boldsymbol{Z}_1]$; <br> $\boldsymbol{X}_2^k[\boldsymbol{Z}_2] = \alpha_2^k \boldsymbol{X}_2^k[\boldsymbol{Z}_2]$; |

We show in Theorem 1 that when $\rho$ is properly chosen, these two formulations lead to the same optimum. See appendix for the complete proof. More importantly, the new formulation 5 allows us to handle the non-negativity constraints and nuclear norm regularization terms separately.

**Theorem 1.** *With the condition $\rho \geqslant \rho^*$, the optimal value $\widehat{\mathrm{OPT}}$ of the problem 5 coincides with the optimal value $\mathrm{OPT}$ in the problem 4 of interest, where $\rho^*$ is a problem dependent threshold,*

$$\rho^* = \max\left\{\frac{\lambda\left(\|\boldsymbol{\Lambda}_0^*\|_* - \|\boldsymbol{Z}_1^*\|_*\right) + \beta\left(\|\boldsymbol{A}^*\|_* - \|\boldsymbol{Z}_2^*\|_*\right)}{\|\boldsymbol{\Lambda}_0^* - \boldsymbol{Z}_1^*\|_F^2 + \|\boldsymbol{A}^* - \boldsymbol{Z}_2^*\|_F^2}\right\}.$$

### 4.3 Efficient Optimization: Proximal Method Meets Conditional Gradient

Now, we are ready to present Algorithm 1 for solving 5 efficiently. Denote $\boldsymbol{X}_1 = [\boldsymbol{\Lambda}_0; \boldsymbol{A}]$, $\boldsymbol{X}_2 = [\boldsymbol{Z}_1; \boldsymbol{Z}_2]$ and $\boldsymbol{X} = [\boldsymbol{X}_1; \boldsymbol{X}_2]$. We use the bracket $[\cdot]$ notation $\boldsymbol{X}_1[\boldsymbol{\Lambda}_0], \boldsymbol{X}_1[\boldsymbol{A}], \boldsymbol{X}_2[\boldsymbol{Z}_1], \boldsymbol{X}_2[\boldsymbol{Z}_2]$ to represent the respective part for simplicity. Let $f(\boldsymbol{X}) := f(\boldsymbol{\Lambda}_0, \boldsymbol{A}, \boldsymbol{Z}_1, \boldsymbol{Z}_2) = -\frac{1}{|\mathcal{O}|}\sum_{\mathcal{T}^{u,i} \in \mathcal{O}} \ell\left(\mathcal{T}^{u,i}|\boldsymbol{\Lambda}_0, \boldsymbol{A}\right) + \rho\|\boldsymbol{\Lambda}_0 - \boldsymbol{Z}_1\|_F^2 + \rho\|\boldsymbol{A} - \boldsymbol{Z}_2\|_F^2$.

The course of our action is straightforward: at each iteration, we apply cheap projection gradient for block $\boldsymbol{X}_1$ and cheap linear minimization for block $\boldsymbol{X}_2$ and maintain three interdependent sequences $\left\{\boldsymbol{U}^k\right\}_{k\geqslant 1}$, $\left\{\boldsymbol{Y}^k\right\}_{k\geqslant 1}$ and $\left\{\boldsymbol{X}^k\right\}_{k\geqslant 1}$ based on the accelerated scheme in [17, 18]. To be more specific, the algorithm consists of two main subroutines:

**Proximal Gradient.** When updating $\boldsymbol{X}_1$, we compute directly the associated proximal operator, which in our case, reduces to the simple projection $\boldsymbol{X}_1^k = \left(\boldsymbol{U}^{k-1} - \eta_k \nabla_1 f(\boldsymbol{U}^{k-1})\right)_+$, where $(\cdot)_+$ simply sets the negative coordinates to zero.

**Conditional Gradient.** When updating $\boldsymbol{X}_2$, instead of computing the proximal operator, we call the linear minimization oracle ($\mathrm{LMO}_\psi$): $\boldsymbol{X}_2^k[\boldsymbol{Z}_1] = \mathrm{argmin}\left\{\langle p_k[\boldsymbol{Z}_1], \boldsymbol{Z}_1\rangle + \psi(\boldsymbol{Z}_1)\right\}$ where $p_k = \nabla_2(f(\boldsymbol{U}^{k-1}))$ is the partial derivative with respect to $\boldsymbol{X}_2$ and $\psi(\boldsymbol{Z}_1) = \lambda\|\boldsymbol{Z}_1\|_*$. We do similar updates for $\boldsymbol{X}_2^k[\boldsymbol{Z}_2]$. The overall performance clearly depends on the efficiency of this LMO, which can be solved efficiently in our case as illustrated in Algorithm 3. Following [27], the linear minimization for our situation requires only : **(i)** computing $\boldsymbol{X}_2^k[\boldsymbol{Z}_1] = \mathrm{argmin}_{\|\boldsymbol{Z}_1\|_*\leqslant 1} \langle p_k[\boldsymbol{Z}_1], \boldsymbol{Z}_1\rangle$, where the minimizer is readily given by $\boldsymbol{X}_2^k[\boldsymbol{Z}_1] = u_1 v_1^\top$, and $u_1, v_1$ are the top singular vectors of $-p_k[\boldsymbol{Z}_1]$; and **(ii)** conducting a line-search that produces a scaling factor $\alpha_1^k = \mathrm{argmin}_{\alpha_1 \geqslant 0} h(\alpha_1)$

$$h(\alpha_1) := \rho\|\boldsymbol{Y}_1^{k-1}[\boldsymbol{\Lambda}_0] - (1-\delta^k)\boldsymbol{Y}_2^{k-1}[\boldsymbol{Z}_1] - \delta^k(\alpha_1 \boldsymbol{X}_2^k[\boldsymbol{Z}_1])\|_F^2 + \lambda\delta^k\alpha_1 + C, \qquad (6)$$

where $C = \lambda(1-\delta^k)\|\boldsymbol{Y}_2^{k-1}[\boldsymbol{Z}_1]\|_*$. The quadratic problem (6) admits a closed-form solution and thus can be computed efficiently. We repeat the same process for updating $\alpha_2^k$ accordingly.

### 4.4 Convergence Analysis

Denote $F(\boldsymbol{X}) = f(\boldsymbol{X}) + \psi(\boldsymbol{X}_2)$ as the objective in formulation 5, where $\boldsymbol{X} = [\boldsymbol{X}_1; \boldsymbol{X}_2]$. We establish the following convergence results for Algorithm 1 described above when solving formulation 5. Please refer to Appendix for complete proof.

**Theorem 2.** *Let $\left\{\boldsymbol{Y}^k\right\}$ be the sequence generated by Algorithm 1 by setting $\delta^k = 2/(k+1)$, and $\eta^k = (\delta^k)^{-1}/L$. Then for $k \geqslant 1$, we have*

$$F(\boldsymbol{Y}^k) - \widehat{\text{OPT}} \leqslant \frac{4LD_1}{k(k+1)} + \frac{2LD_2}{k+1}. \tag{7}$$

*where $L$ corresponds to the Lipschitz constant of $\nabla f(\boldsymbol{X})$ and $D_1$ and $D_2$ are some problem dependent constants.*

**Remark.** Let $g(\Lambda_0, A)$ denote the objective in formulation 4, which is the original problem of our interest. By invoking Theorem 1, we further have, $g(\boldsymbol{Y}^k[\Lambda_0], \boldsymbol{Y}^k[A]) - \text{OPT} \leqslant \frac{4LD_1}{k(k+1)} + \frac{2LD_2}{k+1}$. The analysis builds upon the recursions from proximal gradient and conditional gradient methods. As a result, the overall convergence rate comes from two parts, as reflected in (7). Interestingly, one can easily see that for both the proximal and the conditional gradient parts, we achieve the respective *optimal* convergence rates. When there is no nuclear norm regularization term, the results recover the well-known optimal $O(1/t^2)$ rate achieved by proximal gradient method for smooth convex optimization. When there is no nonnegative constraint, the results recover the well-known $O(1/t)$ rate attained by conditional gradient method for smooth convex minimization. When both nuclear norm and non-negativity are in present, the proposed algorithm, up to our knowledge, is first of its kind, that achieves the best of both worlds, which could be of independent interest.

## 5 Experiments

We evaluate our algorithm by comparing with state-of-the-art competitors on both synthetic and real datasets. For each user, we randomly pick 20-percent of all the items she has consumed and hold out the *entire* sequence of events. Besides, for each sequence of the other 80-percent items, we further split it into a pair of training/testing subsequences. For each testing event, we evaluate the predictive accuracy on two tasks :

(a) **Item Recommendation**: suppose the testing event belongs to the user-item pair $(u, i)$. Ideally item $i$ should rank top at the testing moment. We record its predicted rank among all items. Smaller value indicates better performance.

(b) **Returning-Time Prediction:** we predict the returning-time from the learned intensity function and compute the absolute error with respect to the true time.

We repeat these two evaluations on all testing events. Because the predictive tasks on those entirely held-out sequences are much more challenging, we report the *total* mean absolute error (MAE) and that specific to the set of entirely *heldout* sequences, separately.

### 5.1 Competitors

**Poisson process** is a relaxation of our model by assuming each user-item pair $(u, i)$ has only a constant base intensity $\boldsymbol{\Lambda}_0(u, i)$, regardless of the history. For task (a), it gives static ranks regardless of the time. For task (b), it produces an estimate of the average inter-event gaps. In many cases, the Poisson process is a hard baseline in that the most popular items often have large base intensity, and recommending popular items is often a strong heuristic.

**STiC** [11] fits a semi-hidden Markov model to each observed user-item pair. Since it can only make recommendations specific to the few observed items visited before, instead of the large number of new items, we only evaluate its performance on the returning time prediction task. For the set of entirely held-out sequences, we use the average predicted inter-event time from each observed item as the final prediction.

**SVD** is the classic matrix factorization model. The implicit user feedback is converted into an explicit rating using the frequency of item consumptions [2]. Since it is not designed for predicting the returning time, we report its performance on the time-sensitive recommendation task as a reference.

**Tensor factorization** generalizes matrix factorization to include time. We compare with the state-of-art method [3] which considers poisson regression as the loss function to fit the number of events in each discretized time slot and shows better performance compared to other alternatives with the squared loss [25, 13, 22, 21]. We report the performance by (1) using the parameters fitted only in the last interval, and (2) using the average parameters over all time intervals. We denote these two variants with varying number of intervals as *Tensor-#-Last* and *Tensor-#-Avg*.

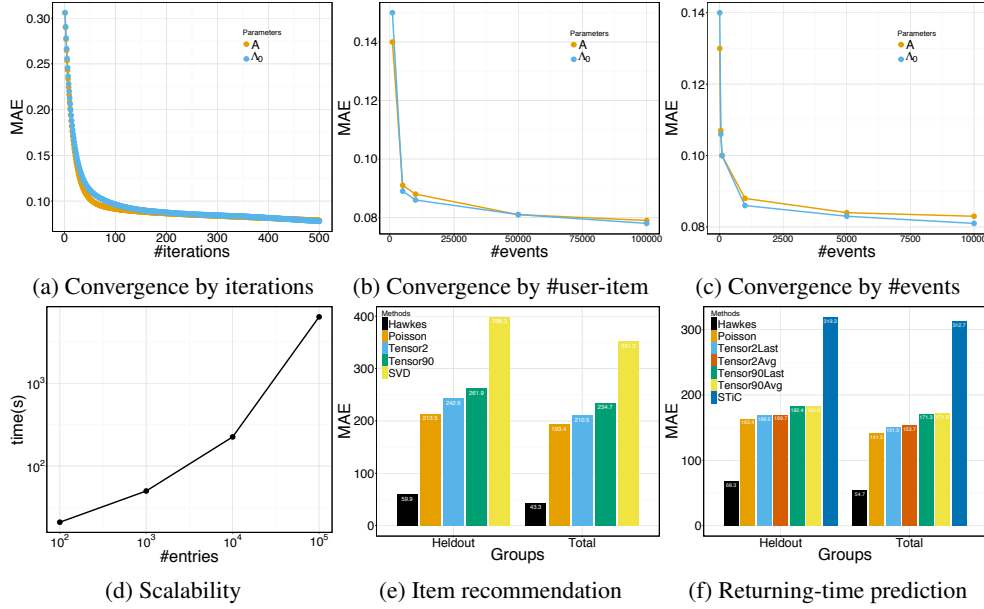

| (a) Convergence by iterations | (b) Convergence by #user-item | (c) Convergence by #events |
| --- | --- | --- |
| (d) Scalability | (e) Item recommendation | (f) Returning-time prediction |

Figure 2: Estimation error (a) by #iterations, (b) by #entries (1,000 events per entry), and (c) by #events per entry (10,000 entries); (d) scalability by #entries (1,000 events per entry, 500 iterations); (e) MAE of the predicted ranking; and (f) MAE of the predicted returning time.

## 5.2 Results

**Synthetic data.** We generate two 1,024-by-1,204 user-item matrices $\Lambda_0$ and $A$ with rank five as the ground-truth. For each user-item pair, we simulate 1,000 events by Ogata's thinning algorithm [19] with an exponential triggering kernel and get 100 million events in total. The bandwidth for the triggering kernel is fixed to one. By theorem 1, it is inefficient to directly estimate the exact value of the threshold value for $\rho$. Instead, we tune $\rho$, $\lambda$ and $\beta$ to give the best performance.

*How does our algorithm converge ?* Figure 2(a) shows that it only requires a few hundred iterations to descend to a decent error for both $\Lambda_0$ and $A$, indicating algorithm 1 converges very fast. Since the true parameters are low-rank, Figure 2(b-c) verify that it only requires a modest number of observed entries, each of which induces a small number of events (1,000) to achieve a good estimation performance. Figure 2(d) further illustrates that algorithm 1 scales linearly as the training set grows.

*What is the predictive performance ?* Figure 2(e-f) confirm that algorithm 1 achieves the best predictive performance compared to other baselines. In Figure 2(e), all temporal methods outperform the static SVD since this classic baseline does not consider the underlying temporal dynamics of the observed sequences. In contrast, although the Poisson regression also produces static rankings of the items, it is equivalent to recommending the most popular items over time. This simple heuristic can still give competitive performance. In Figure 2(f), since the occurrence of a new event depends on the whole past history instead of the last one, the performance of STiC deteriorates vastly. The other tensor methods predict the returning time with the information from different time intervals. However, because our method automatically adapts different contributions of each past event to the prediction of the next event, it can achieve the best prediction performance overall.

**Real data.** We also evaluate the proposed method on real datasets. *last.fm* consists of the music streaming logs between 1,000 users and 3,000 artists. There are around 20,000 observed user-artist pairs with more than one million events in total. *tmall.com* contains around 100K shopping events between 26,376 users and 2,563 stores. The unit time for both dataset is hour. MIMIC II medical dataset is a collection of de-identified clinical visit records of Intensive Care Unit patients for seven years. We filtered out 650 patients and 204 diseases. Each event records the time when a patient was diagnosed with a specific disease. The time unit is week. All model parameters $\rho$, $\lambda$, $\beta$, the kernel bandwidth and the latent rank of other baselines are tuned to give the best performance.

*Does the history help ?* Because the true temporal dynamics governing the event patterns are unobserved, we first investigate whether our model assumption is reasonable. Our *Hawkes* model considers the self-exciting effects from past user activities, while the survival analysis applied in [11]

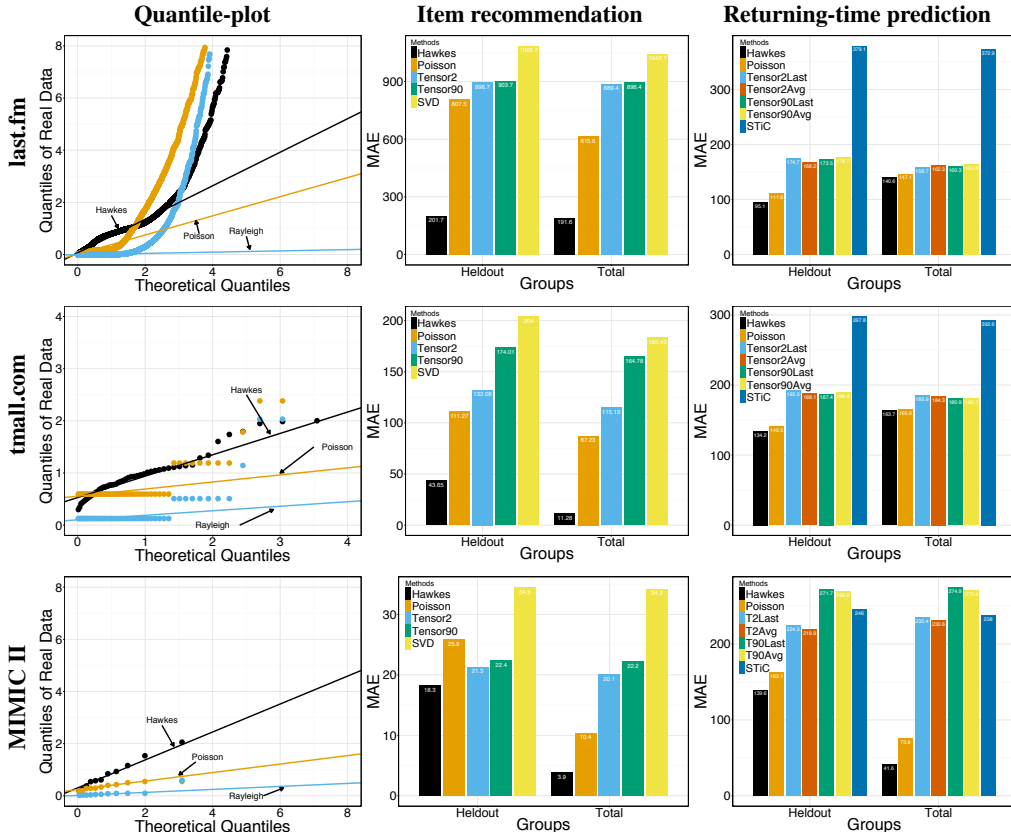

Figure 3: The quantile plots of different fitted processes, the MAE of predicted rankings and returning-time on the *last.fm* (top), *tmall.com* (middle) and the MIMIC II (bottom), respectively.

assumes *i.i.d.* inter-event gaps which might conform to an exponential (Poisson process) or Rayleigh distribution. According to the time-change theorem [6], given a sequence $\mathcal{T} = \{t_1, \ldots, t_n\}$ and a particular point process with intensity $\lambda(t)$, the set of samples $\left\{\int_{t_i-1}^{t_i} \lambda(t)dt\right\}_{i=1}^{n}$ should conform to a unit-rate exponential distribution if $\mathcal{T}$ is truly sampled from the process. Therefore, we compare the theoretical quantiles from the exponential distribution with the fittings of different models to a real sequence of (listening/shopping/visiting) events. The closer the slope goes to one, the better a model matches the event patterns. Figure 3 clearly shows that our *Hawkes* model can better explain the observed data compared to the other survival analysis models.

*What is the predictive performance ?* Finally, we evaluate the prediction accuracy in the $2nd$ and $3rd$ column of Figure 3. Since holding-out an entire testing sequence is more challenging, the performance on the *Heldout* group is a little lower than that on the average *Total* group. However, across all cases, since the proposed model is able to better capture the temporal dynamics of the observed sequences of events, it can achieve a better performance on both tasks in the end.

## 6   Conclusions

We propose a novel convex formulation and an efficient learning algorithm to recommend relevant services at any given moment, and to predict the next returning-time of users to existing services. Empirical evaluations on large synthetic and real data demonstrate its superior scalability and predictive performance. Moreover, our optimization algorithm can be used for solving general nonnegative matrix rank minimization problem with other convex losses under mild assumptions, which may be of independent interest.

## Acknowledge

The research was supported in part by NSF IIS-1116886, NSF/NIH BIGDATA 1R01GM108341, NSF CAREER IIS-1350983.

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
