[Supplementary Material]

# 7 Appendix

**Theorem 1.** *With the condition $\rho \geqslant \rho^*$, the optimal value $\widehat{\mathrm{OPT}}$ of the problem 5 coincides with the optimal value $\mathrm{OPT}$ in the problem 8 of interest, where $\rho^*$ is a problem dependent threshold.*

*Proof.* We start by rewriting formulation 4 to the equivalent form:

$$\min -\frac{1}{|\mathcal{O}|} \sum_{\mathcal{T}^{u,i} \in \mathcal{O}} \ell\left(\mathcal{T}^{u,i}|\boldsymbol{\Lambda}_0, \boldsymbol{A}\right) + \lambda \|\boldsymbol{Z}_1\|_* + \beta \|\boldsymbol{Z}_2\|_*, \ \boldsymbol{\Lambda}_0, \boldsymbol{A} \geqslant \boldsymbol{0}, \boldsymbol{\Lambda}_0 = \boldsymbol{Z}_1, \boldsymbol{A} = \boldsymbol{Z}_2. \quad (8)$$

We can observe that the optimal solution of 8 is a feasible solution of 5 with the same objective function value, so it is evident that $\widehat{\mathrm{OPT}} \leqslant \mathrm{OPT}$. On the other hand, suppose $(\boldsymbol{\Lambda}_0^*, \boldsymbol{A}^*, \boldsymbol{Z}_1^*, \boldsymbol{Z}_2^*)$ is an optimal solution of 5 with $\boldsymbol{\Lambda}_0^* \neq \boldsymbol{Z}_1^*, \boldsymbol{A}^* \neq \boldsymbol{Z}_2^*$ in general. Since $\boldsymbol{\Lambda}_0^*, \boldsymbol{A}^* \geqslant \boldsymbol{0}$, they are also feasible for 8, so we can find a $\rho'$ such that $\lambda \|\boldsymbol{Z}_1^*\|_* + \beta \|\boldsymbol{Z}_2^*\|_* + \rho' \|\boldsymbol{\Lambda}_0^* - \boldsymbol{Z}_1^*\|_F^2 + \rho' \|\boldsymbol{A}^* - \boldsymbol{Z}_2^*\|_F^2 \geqslant \lambda \|\boldsymbol{\Lambda}_0^*\|_* + \beta \|\boldsymbol{A}^*\|_*$. Therefore, under the condition that

$$\rho \geqslant \rho^* = \max \left\{ \frac{\lambda \left(\|\boldsymbol{\Lambda}_0^*\|_* - \|\boldsymbol{Z}_1^*\|_*\right) + \beta \left(\|\boldsymbol{A}^*\|_* - \|\boldsymbol{Z}_2^*\|_*\right)}{\|\boldsymbol{\Lambda}_0^* - \boldsymbol{Z}_1^*\|_F^2 + \|\boldsymbol{A}^* - \boldsymbol{Z}_2^*\|_F^2} \right\}, \quad (9)$$

we have $\widehat{\mathrm{OPT}} \geqslant -\frac{1}{|\mathcal{O}|} \sum_{\mathcal{T}^{u,i} \in \mathcal{O}} \ell\left(\mathcal{T}^{u,i}|\boldsymbol{\Lambda}_0^*, \boldsymbol{A}^*\right) + \lambda \|\boldsymbol{\Lambda}_0^*\|_* + \beta \|\boldsymbol{A}^*\|_* \geqslant \mathrm{OPT}$ and readily arrive at the theorem. $\square$

**Theorem 2.** *Let $\left\{\boldsymbol{Y}^k\right\}$ be the sequence generated by Algorithm 1, $\delta^k = 2/(k+1)$, and $\eta^k = (\delta^k)^{-1}/L$, $D_1$ and $D_2$ some problem dependent constants. Then for $k \geqslant 1$, we have*

$$F(\boldsymbol{Y}^k) - F(\boldsymbol{X}^*) \leqslant \frac{4LD_1}{k(k+1)} + \frac{2LD_2}{k+1}. \quad (10)$$

*Proof.* Consider the following general optimization problem

$$\min_{\boldsymbol{X} \in \Omega} F(\boldsymbol{X}) := f(\boldsymbol{X}_1; \boldsymbol{X}_2) + \Psi(\boldsymbol{X}_2), \quad (11)$$

where $\boldsymbol{X} = [\boldsymbol{X}_1; \boldsymbol{X}_2]$, $\Omega = \Omega_1 \times \Omega_2$, $f$ is $L$-smooth and convex, and $\Psi(\cdot)$ is convex. Let $\delta^k = \frac{1}{k+2}$ and $\eta^k = (\delta^k)^{-1}/L$. First Note that $\boldsymbol{Y}^k - \boldsymbol{U}^{k-1} = \delta^k (\boldsymbol{X}^k - \boldsymbol{X}^{k-1})$. By the smoothness of $f$ where $f(y) \leqslant f(x) + \langle f'(x), y - x \rangle + \frac{L}{2} \|y - x\|^2$, we have

$$
\begin{aligned}
f(\boldsymbol{Y}^k) \ &\leqslant \ f(\boldsymbol{U}^{k-1}) + \nabla f(\boldsymbol{U}^{k-1})^\top \left(\boldsymbol{Y}^k - \boldsymbol{U}^{k-1}\right) + \tfrac{L}{2} \delta_k^2 \|\boldsymbol{X}_k - \boldsymbol{X}_{k-1}\|^2 \\
&\qquad \left(\text{by the definition of } \boldsymbol{Y}^k\right) \\
&= \ (1 - \delta^k)\left(f(\boldsymbol{U}^{k-1}) + \nabla f(\boldsymbol{U}^{k-1})^\top \left(\boldsymbol{Y}^{k-1} - \boldsymbol{U}^{k-1}\right)\right) \\
&\qquad + \delta^k \left(f(\boldsymbol{U}^{k-1}) + \nabla f(\boldsymbol{U}^{k-1})^\top \left(\boldsymbol{X}^k - \boldsymbol{U}^{k-1}\right)\right) + \tfrac{L}{2} \delta_k^2 \|\boldsymbol{X}_k - \boldsymbol{X}_{k-1}\|^2 \quad (12) \\
&\qquad \left(\text{by the convexity of } f\right) \\
&\leqslant \ (1 - \delta^k) f(\boldsymbol{Y}^{k-1}) + \delta^k \left(f(\boldsymbol{U}^{k-1}) + \nabla f(\boldsymbol{U}^{k-1})^\top \left(\boldsymbol{X}^k - \boldsymbol{U}^{k-1}\right)\right) \\
&\qquad + \tfrac{L}{2} \delta_k^2 \|\boldsymbol{X}_k - \boldsymbol{X}_{k-1}\|^2.
\end{aligned}
$$

Note the proximal mapping $\mathrm{Prox}_{x_0}(\xi) := \mathrm{argmin}_{x \in X} \{V(x, x_0) + \langle \xi, x \rangle\}$, where $V(x, x') = \omega(x) - \omega(x') - \langle \nabla \omega(x'), x - x' \rangle$ is the Bregman distance, and $\omega(x)$ is 1-strongly convex. For any $\boldsymbol{X}_1 \in \Omega_1$, we have the following well-known inequality [20] :

$$\nabla_1 f(\boldsymbol{U}^{k-1})^T (\boldsymbol{X}_1^k - \boldsymbol{X}_1) \leqslant (\eta^k)^{-1}[V(\boldsymbol{X}_1, \boldsymbol{X}_1^{k-1}) - V(\boldsymbol{X}_1, \boldsymbol{X}_1^k) - V(\boldsymbol{X}_1^k, \boldsymbol{X}_1^{k-1})]. \quad (13)$$

Besides, by our linear minimization oracle

$$\mathrm{LMO}_\Psi(\nabla_2 f(\boldsymbol{U}^{k-1})) = \mathrm{argmin} \left\{\langle \nabla_2 f(\boldsymbol{U}^{k-1}), \boldsymbol{X}_2 \rangle + \Psi(\boldsymbol{X}_2)\right\}, \quad (14)$$

we have

$$\nabla_2 f(\boldsymbol{U}^{k-1})^\top \boldsymbol{X}_2^k + \Psi(\boldsymbol{X}_2^k) \leqslant \nabla_2 f(\boldsymbol{U}^{k-1})^\top \boldsymbol{X}_2 + \Psi(\boldsymbol{X}_2). \quad (15)$$

As a consequence,

$$\delta^k \left( f(\boldsymbol{U}^{k-1}) + \nabla f(\boldsymbol{U}^{k-1})^\top \left( \boldsymbol{X}^k - \boldsymbol{U}^{k-1} \right) \right)$$

$$= \delta^k \left( f(\boldsymbol{U}^{k-1}) + \nabla_1 f(\boldsymbol{U}^{k-1})^\top \left( \boldsymbol{X}_1^k - \boldsymbol{U}_1^{k-1} \right) + \nabla_2 f(\boldsymbol{U}^{k-1})^\top \left( \boldsymbol{X}_2^k - \boldsymbol{U}_2^{k-1} \right) \right)$$

(by equation 15)

$$\leqslant \delta^k ( f(\boldsymbol{U}^{k-1}) + \nabla_1 f(\boldsymbol{U}^{k-1})^\top \left( \boldsymbol{X}_1^k - \boldsymbol{U}_1^{k-1} + \boldsymbol{X}_1^* - \boldsymbol{X}_1^* \right)$$
$$+ \nabla_2 f(\boldsymbol{U}^{k-1})^\top \boldsymbol{X}_2^* + \Psi(\boldsymbol{X}_2^*) - \Psi(\boldsymbol{X}_2^k) - \nabla_2 f(\boldsymbol{U}^{k-1})^\top \boldsymbol{U}_2^{k-1} ))$$
$$\leqslant \delta^k ( f(\boldsymbol{U}^{k-1}) + \nabla_1 f(\boldsymbol{U}^{k-1})^\top \left( \boldsymbol{X}_1^* - \boldsymbol{U}_1^{k-1} \right) + \nabla_2 f(\boldsymbol{U}^{k-1})^\top (\boldsymbol{X}_2^* - \boldsymbol{U}_2^{k-1}) + \Psi(\boldsymbol{X}_2^*)$$
$$+ \nabla_1 f(\boldsymbol{U}^{k-1})^\top \left( \boldsymbol{X}_1^k - \boldsymbol{X}_1^* \right) - \Psi(\boldsymbol{X}_2^k) )$$

(by the convexity of $f$)

$$\leqslant \delta^k F(\boldsymbol{X}^*) + \delta^k \nabla_1 f(\boldsymbol{U}^{k-1})^\top \left( \boldsymbol{X}_1^k - \boldsymbol{X}_1^* \right) - \delta^k \Psi(\boldsymbol{X}_2^k)$$

(by equation 13)

$$\leqslant \delta^k F(\boldsymbol{X}^*) + \delta^k (\eta^k)^{-1}(V(\boldsymbol{X}_1^*, \boldsymbol{X}_1^{k-1}) - V(\boldsymbol{X}_1^*, \boldsymbol{X}_1^k) - V(\boldsymbol{X}_1^k, \boldsymbol{X}_1^{k-1})) - \delta^k \Psi(\boldsymbol{X}_2^k)$$

(by the definition of Bregman distance)

$$\leqslant \delta^k F(\boldsymbol{X}^*) + L(\delta^k)^2(V(\boldsymbol{X}_1^*, \boldsymbol{X}_1^{k-1}) - V(\boldsymbol{X}_1^*, \boldsymbol{X}_1^k)) - \frac{L(\delta^k)^2}{2} \| \boldsymbol{X}_1^k - \boldsymbol{X}_1^{k-1} \|^2 - \delta^k \Psi(\boldsymbol{X}_2^k)$$

Plugging into the previous inequality 12, we end up with

$$f(\boldsymbol{Y}^k) \leqslant (1 - \delta^k) f(\boldsymbol{Y}^{k-1}) + \delta^k F(\boldsymbol{X}^*) + L(\delta^k)^2 (V(\boldsymbol{X}_1^*, \boldsymbol{X}_1^{k-1}) - V(\boldsymbol{X}_1^*, \boldsymbol{X}_1^k))$$
$$+ \frac{L(\delta^k)^2}{2} \| \boldsymbol{X}_2^k - \boldsymbol{X}_2^{k-1} \|^2 - \delta^k \Psi(\boldsymbol{X}_2^k), \tag{16}$$

where we have used the fact $\| \boldsymbol{X} = (\boldsymbol{X}_1; \boldsymbol{X}_2) \|^2 = \| \boldsymbol{X}_1 \|^2 + \| \boldsymbol{X}_2 \|^2$. Adding $\Psi(\boldsymbol{Y}_2^k)$ to the both sides, we have

$$F(\boldsymbol{Y}^k) \leqslant (1 - \delta^k) F(\boldsymbol{Y}^{k-1}) + \delta^k F(\boldsymbol{X}^*) + L(\delta^k)^2 (V(\boldsymbol{X}_1^*, \boldsymbol{X}_1^{k-1}) - V(\boldsymbol{X}_1^*, \boldsymbol{X}_1^k))$$
$$+ \frac{L(\delta^k)^2}{2} \| \boldsymbol{X}_2^k - \boldsymbol{X}_2^{k-1} \|^2 + \Psi(\boldsymbol{Y}_2^k) - \delta^k \Psi(\boldsymbol{X}_2^k) - (1 - \delta^k) \Psi(\boldsymbol{Y}_2^{k-1})$$

$$\left( \text{by the convexity of } \Psi \text{ and the definition of } \boldsymbol{Y}^k \right) \tag{17}$$

$$\leqslant (1 - \delta^k) F(\boldsymbol{Y}^{k-1}) + \delta^k F(\boldsymbol{X}^*) + L(\delta^k)^2 (V(\boldsymbol{X}_1^*, \boldsymbol{X}_1^{k-1}) - V(\boldsymbol{X}_1^*, \boldsymbol{X}_1^k))$$
$$+ \frac{L(\delta^k)^2}{2} \| \boldsymbol{X}_2^k - \boldsymbol{X}_2^{k-1} \|^2.$$

Subtracting $F(\boldsymbol{X}^*)$ from both sides of the above inequality, we have

$$F(\boldsymbol{Y}^k) - F(\boldsymbol{X}^*) \leqslant (1 - \delta^k)(F(\boldsymbol{Y}^{k-1}) - F(\boldsymbol{X}^*)) + L(\delta^k)^2 (V(\boldsymbol{X}_1^*, \boldsymbol{X}_1^{k-1}) - V(\boldsymbol{X}_1^*, \boldsymbol{X}_1^k))$$
$$+ \frac{L(\delta^k)^2}{2} \| \boldsymbol{X}_2^k - \boldsymbol{X}_2^{k-1} \|^2. \tag{18}$$

By the fact $\delta^1 = 1$ and invoking the Lemma 1 of [18], the above inequality implies that

$$F(\boldsymbol{Y}^k) - F(\boldsymbol{X}^*) \leqslant \frac{4L}{k(k+1)} \left( V(\boldsymbol{X}_1^*, \boldsymbol{X}_1^0) + \frac{1}{2} \sum_{i=1}^{k} \| \boldsymbol{X}_2^k - \boldsymbol{X}_2^{k-1} \|^2 \right). \tag{19}$$

Let $D_1 = V(\boldsymbol{X}_1^*, \boldsymbol{X}_1^0) \geqslant 0$ and $D_2 = \max_{x,y \in \Omega_2} \| x - y \|^2$, we have

$$F(\boldsymbol{Y}^k) - F(\boldsymbol{X}^*) \leqslant \frac{4LD_1}{k(k+1)} + \frac{2LD_2}{k+1}. \tag{20}$$

$\square$