[Reviews · NeurIPS 2015]

Submitted by Assigned_Reviewer_1

Paper proposes a new method for time-sensitive recommendations based on user activities. It is different from existing methods because it address the problem of time-varying user preferences. The paper also addresses prediction of the next returning time of a user. Learning is performed using an efficient optimization algorithm proposed by authors. Experimental section shows results on one synthetic data set in order to show that learning is efficient on large scale data and on two real data sets of modest size.

First, is a problem of time-varying user preferences really that spread. It would be great to analyze the real data sets from experiments in order to make this claim. If so, is it varying mostly because new popular songs come out? Or there some other pattern?

What is the motivation for predicting next returning time for user? Could you please motivate it before you claim it as contribution of a paper? How could a web-company use this prediction and for what?

I believe there are some simple baselines for predicting the next returning time that were not tested: 1) the last time delay 2) the most occurred time delay. Also, I think that predicting the actual returning time of the user is too challenging. Instead, I would focus on a more simpler task that is easier to predict. For example predicting is user will have short delay (1-5 days), medium delay (5-14 days), long delay (14-33 days) or very long delay (>33 days) before he uses our service again.

Experimental section: It was hard for me to get a sense of how significant the performance lift is compared to SVD. Is it groundbreaking (order of magnitude) or modest? Would the user feel the difference? How would these recommendations compare to recommending popular items? In many cases it is a hard baseline to beat, especially if popular items are re-calculated every day.

Small comments:

1) Typo in last paragraph on page 3: "and is abel to capture", abel -> able

2) In first paragraph of section 2 you start mentioning retweets and link creation in social networks without any references. The change of context was confusing.. best to rephrase.
Summary: Paper addresses two problems: time-sensitive recommendations and returning time of user, without any real motivation on why these problems are relevant and how would solving it have an impact. Several baselines and analysis are missing (details bellow). Other than that, overall OK/marginal paper.

Submitted by Assigned_Reviewer_2

-- Summary -- This paper proposed a point process collaborative filtering model to capture the temporal and recurrent structure between users and items. Specifically, for each pair of user and item, the event density is the sum of a constant intensity and a self-exciting intensity depending on the past events. Two sets of intensity are computed by two sets of user and item latent vectors respectively. The corresponding optimization algorithm is derived for parameter estimation and empirical studies have been done to demonstrate the advantage of the proposed methods to other commonly used alternatives.

However, the experimental studies can be improved. For example, another sensible baseline would be a simple ensemble of STiC [12] and SVD. Since the authors argued that [12, 13] are not able to recommend completely new items, which can be covered by SVD, and it would be valuable to compare the proposed method to this simple ensemble. In addition, for the \gamma function used for the self-exciting process, how is the bandwidth parameter \sigma set? How sensitive would the results be if we vary \sigma? Furthermore, there is no details about how some other important hyperparameters are chosen for both baseline methods and the proposed method, e.g. the dimension of the latent vectors.

It would also be helpful to provide more details of the characteristics of the two real world dataset. Since the paper is about temporal structure, it's useful to discuss about simple stats like the number of recurring events per user/item pair and how irregular the temporal pattern is.

-- Quality -- The overall quality is reasonably good but not great.

-- Clarity -- The paper presentation can be improved in terms of languages, typos, grammar errors.

-- Originality --

The proposed method is an interesting combination of point process and latent factor models.

-- Significance --

Both the problem and the method have good practical values.
Summary: This paper proposed a point process collaborative filtering model to capture the temporal and recurrent structure between users and items. The corresponding optimization algorithm is derived for parameter estimation and empirical studies have been done to demonstrate the advantage of the proposed methods to other commonly used alternatives. However, the experimental studies and the presentation can be improved.

Submitted by Assigned_Reviewer_3

This paper presents a novel algorithm to tackle time-sensitive recommendation by capturing low-rank structure in a user-item matrix and by using Hawkes process. The algorithm can be solved in a distributed fashion and achieves better results than existing methods.

Quality: Overall I think the paper took a principled approach to model the observation (ratings) using stochastic processes. But I have a few comments.

One concern in theoretical part is in Eq. (8) and Theorem 2. In eq. 8, over what parameter do we take maximum? Is it over \rho? In such case, can \rho* be unbounded? In addition, theorem 2 is not used when the paper chose \rho in experiments, which degrades the importance of the theorem.

Another comment is with the baselines in the experiments. To my knowledge, SVD++ [17] (the winner of Netflix challenge) is one of the best algorithm for time-sensitive algorithm. Adding SVD++ into baseline would improve the quality even better. Though I am not dissatisfied with the current baselines.

Clarity: The paper was well written and easy to follow.

Originality: To me the paper seems to be novel because time-sensitive recommendation is a relatively novel problem.

Significance: I think time-sensitive recommendation can be impactful.
Summary: This paper presents a novel algorithm to tackle time-sensitive recommendation by capturing low-rank structure in a user-item matrix and by using Hawkes process. The algorithm can be solved in a distributed fashion and achieves better results than existing methods.

Author Feedback
Author rebuttal: We'd like to thank the reviewers for their careful readings and valuable comments. We believe the constructive feedback will improve the paper and increase its potential impact to the community.

First, we'd like to emphasize the contributions:

1. We propose a novel convex formulation for time-sensitive recommendations and returning-time predictions by establishing an under explored connection between self-exciting point processes and low-rank models.
2. We develop an efficient optimization algorithm scaling up to thousands of millions of user-item visiting events.
3. We achieve better predictive performance compared to other state-of-the-arts on both synthetic and real data.
4. The proposed method can be readily generalized to incorporate other important contextual information by making the model explicitly dependent on the additional spatial, textual, categorical, and user profile information.

Reviewer 1

We hope to clarify the motivations for making time-sensitive recommendations and predicting the returning-time. Although the following important applications are from different domains, they can be well captured by the proposed model:

I. As a web company, like Google and Facebook, time-sensitive recommendations can have potential impact first to display ads. If we can predict when our users will come back next, we can make the existing ads bidding much more economic, allowing marketers to bid on time slots. After all, marketers do not need to blindly bid all time slots indiscriminately.
II. For most on-line stores, accurate prediction of the returning time of customers can help to improve stock management and products display and arrangement.
III. For mainstream personal assistants, like Google Now, because people tend to have different activities dependent on the temporal/spatial contexts like morning vs. evening, weekdays vs. weekend, making recommendations on the right thing and at the right moment can make such services more relevant and usable.
IV. In modern electronic health record data, patients may have several diseases that have complicated dependencies on each other. The occurrence of one disease can trigger the progression of others. Predicting the returning time on certain disease can effectively help to take proactive steps to reduce the potential risks.

As for comparing to simple baselines, the competitor Tensor2Last and Tensor90Last in the paper tend to utilize the last time delay, and the respective Tensor2Avg, Tensor90Avg and Poisson focus on the average time delay instead. Since the time is continuous, most frequent delay can be approximated by the average in general.
As for recommending most popular item, that's equivalent to recommend the item with largest base intensity (or Poisson regression).

Predicting user behaviors in discrete-time buckets will be useful in practice. Yet, given an increasing amount of data (number of users, length of history, increasing detail of user profile), it is likely we will be able to predict the returning time more and more accurately. Our proposed method is targeting at such trend in modern data collection.

Because SVD gives static rankings from predicted ratings but cannot make recommendations and predictions in the future, the relatively new setting we are working with is groundbreaking. Since the recommended items are different by the temporal contexts, users can see the difference when they log in each time. Compared to recommending popular items each day, because people may go to parks, cinemas, restaurants during a weekend, our proposed model can also capture these repeated visiting patterns per user, leading to improved personalization.

Reviewer 2 and 4

In a sense the ensemble of STiC and SVD is an adhoc combination of temporal point processes and low rank models. In comparison, our proposed low rank Hawkes process is a principled integration of point processes and low rank models, which we believe will do better. We can add such empirical comparison in the paper. The bandwidth for the triggering kernel and the dimension of the latent vectors are chosen using cross-validation to give the best results. We do have more details of the characteristics of real world data, and will bring them back in the final version by improving the organization of the paper.

Reviewer 3

From theorem 2, it is hard to directly estimate the exact value of the threshold value for \rho. Instead, we empirically tune \rho staring from 1 using cross-validations to find the best value. SVD++ weights more recent user-item ratings than old ones to model the most recent user preferences. Similar to the most recent Tensor methods compared in our experiments, SVD++ cannot extrapolate beyond a predefined observation window, and their performance depends on how we partition the time into intervals. Moreover, it does not make any predictions of users' future returning time. We can add the comparison in the final version of our paper.